# Pharmacological brake-release of mRNA translation enhances cognitive memory

**Carmela Sidrauski[1,2]\***, **Diego Acosta-Alvear[1,2]**, **Arkady Khoutorsky[3]**, **Punitha Vedantham[4]**, **Brian R Hearn[4]**, **Han Li[5]**, **Karine Gamache[6]**, **Ciara M Gallagher[1,2]**, **Kenny K-H Ang[4]**, **Chris Wilson[4]**, **Voytek Okreglak[1,2]**, **Avi Ashkenazi[5]**, **Byron Hann[7]**, **Karim Nader[6]**, **Michelle R Arkin[4]**, **Adam R Renslo[4]**, **Nahum Sonenberg[3]**, **Peter Walter[1,2]\***

[1]Department of Biochemistry and Biophysics, University of California, San Francisco, San Francisco, United States; [2]Howard Hughes Medical Institute, University of California, San Francisco, San Francisco, United States; [3]Department of Biochemistry, McGill Cancer Center, Montreal, Canada; [4]Small Molecule Discovery Center and Department of Pharmaceutical Chemistry, University of California, San Francisco, San Francisco, United States; [5]Department of Molecular Oncology, Genentech Inc, South San Francisco, United States; [6]Department of Psychology, McGill University, Montreal, Canada; [7]Helen Diller Family Comprehensive Cancer Center, University of California, San Francisco, San Francisco, United States

**Abstract** Phosphorylation of the α-subunit of initiation factor 2 (eIF2) controls protein synthesis by a conserved mechanism. In metazoa, distinct stress conditions activate different eIF2α kinases (PERK, PKR, GCN2, and HRI) that converge on phosphorylating a unique serine in eIF2α. This collection of signaling pathways is termed the 'integrated stress response' (ISR). eIF2α phosphorylation diminishes protein synthesis, while allowing preferential translation of some mRNAs. Starting with a cell-based screen for inhibitors of PERK signaling, we identified a small molecule, named ISRIB, that potently (IC$_{50}$ = 5 nM) reverses the effects of eIF2α phosphorylation. ISRIB reduces the viability of cells subjected to PERK-activation by chronic endoplasmic reticulum stress. eIF2α phosphorylation is implicated in memory consolidation. Remarkably, ISRIB-treated mice display significant enhancement in spatial and fear-associated learning. Thus, memory consolidation is inherently limited by the ISR, and ISRIB releases this brake. As such, ISRIB promises to contribute to our understanding and treatment of cognitive disorders.

**\*For correspondence:**
carmelas@me.com (CS);
peter@walterlab.ucsf.edu (PW)

**Competing interests:**
See page 20

**Reviewing editor**: David Ron, Cambridge University, United Kingdom

## Introduction

In metazoa, diverse stress signals converge at a single phosphorylation event at serine 51 of a common effector, the translation initiation factor eIF2α. This step is carried out by four eIF2α kinases in mammalian cells: PERK, which responds to an accumulation of unfolded proteins in the endoplasmic reticulum (ER), GCN2 to amino acid starvation and UV light, PKR to viral infection, and HRI to heme deficiency. This collection of signaling pathways has been termed the 'integrated stress response' (ISR), as they converge on the same molecular event. eIF2α phosphorylation results in an attenuation of translation with consequences that allow cells to cope with the varied stresses (*Wek et al., 2006*).

eIF2 (which is comprised of three subunits, α, β and γ) binds GTP and the initiator Met-tRNA to form the ternary complex (eIF2-GTP-Met-tRNA$_i$), which, in turn, associates with the 40S ribosomal subunit forming the 43S pre-initiation complex (PIC) that scans the 5'UTR of mRNAs to select the initiating AUG codon. Upon phosphorylation of its α-subunit, eIF2 becomes a competitive inhibitor of its

**eLife digest** The synthesis of proteins is an essential step in many biological processes, including memory, and drugs that inhibit protein synthesis are known to impair memory in rodents. It is thought that the brain needs these proteins to convert short-term memories into long-term memories through a process known as consolidation.

A protein called EIF2α has a key role in the regulation of protein synthesis, and has also been implicated in memory. EIF2α can be activated as a result of being phosphorylated by any of four protein kinases: these are in turn activated by processes that subject cells to stress, such as viral infection, UV light or—in the case of a kinase known as PERK—the accumulation of unfolded proteins in a cellular organelle called the endoplasmic reticulum. Activation of EIF2α downregulates most protein synthesis inside the cell, but upregulates the production of a small number of key regulatory molecules: these changes help cells to cope with whatever stressful event they have just experienced.

To obtain further insight into the cellular stress response, Sidrauski et al. screened a large library of compounds in search of one that inhibits PERK. They identified a molecule—known as ISRIB—which acts downstream of all four protein kinases by reversing the effects of EIF2α phosphorylation. ISRIB is the first molecule shown to have this effect, and thus represents an important tool for investigating the stress response inside cells.

When Sidrauski et al. injected ISRIB into mice, the animals showed improved memory: for example, they learnt to locate a hidden platform in a water maze more rapidly than controls. This suggests that ISRIB could be used to explore the mechanisms that underlie memory consolidation, and possibly even as a memory enhancer. Moreover, given that many tumor cells exploit the cellular stress response to aid their own growth, ISRIB may have potential as a novel chemotherapeutic agent.

guanine nucleotide exchange factor (GEF), eIF2B (*Hinnebusch and Lorsch, 2012*). The tight and non-productive binding of phosphorylated eIF2 to eIF2B prevents loading of the eIF2 complex with GTP thus blocking ternary complex formation and reducing translation initiation (*Krishnamoorthy et al., 2001*). Because eIF2B is less abundant than eIF2, phosphorylation of only a small fraction of the total eIF2 has a dramatic impact on eIF2B activity in cells.

Paradoxically, under conditions of reduced protein synthesis, a small group of mRNAs that contain upstream open reading frames (uORFs) in their 5′UTR are translationally up-regulated (*Hinnebusch, 2005*; *Jackson et al., 2010*). These include mammalian ATF4 (a cAMP element binding [CREB] transcription factor) and CHOP (a pro-apoptotic transcription factor) (*Harding et al., 2000*; *Vattem and Wek, 2004*; *Palam et al., 2011*). ATF4 regulates the expression of many genes involved in metabolism and nutrient uptake and additional transcription factors, such as CHOP, which is under both translational and transcriptional control (*Ma et al., 2002*). Phosphorylation of eIF2α thus leads to preferential translation of key regulatory molecules and directs diverse changes in the transcriptome of cells upon cellular stress.

One of the eIF2α kinases, PERK, lies at the intersection of the ISR and the unfolded protein response (UPR) that maintains homeostasis of protein folding in the ER (*Pavitt and Ron, 2012*). The UPR is activated by unfolded or misfolded proteins that accumulate in the ER lumen because of an imbalance between protein folding load and protein folding capacity, a condition known as 'ER stress'. In mammals, the UPR is comprised of three signaling branches mediated by ER-localized transmembrane sensors, PERK, IRE1, and ATF6. These sensor proteins detect the accumulation of unfolded protein in the ER and transmit the information across the ER membrane, initiating unique signaling pathways that converge in the activation of an extensive transcriptional response, which ultimately results in ER expansion (*Ron and Walter, 2007*). The lumenal stress-sensing domains of PERK and IRE1 are homologous and likely activated in analogous ways by direct binding to unfolded peptides (*Gardner and Walter, 2011*). This binding event leads to oligomerization and *trans*-autophosphorylation of their cytosolic kinase domains, and, for PERK, phosphorylation of its only known substrate, eIF2α. In this way, PERK activation results in a quick reduction in the load of newly synthesized proteins that are translocated into the ER-lumen (*Harding et al., 2000*).

Upon ER stress, both the transcription factor XBP1s, produced as the consequence of a non-conventional mRNA splicing reaction initiated by IRE1, and the transcription factor ATF6, produced

by proteolysis and release from the ER membrane, collaborate with ATF4 to induce the vast UPR transcriptional response. Transcriptional targets of the UPR include the ER protein folding machinery, the ER-associated degradation machinery, and many other components functioning in the secretory pathway (*Walter and Ron, 2011*). Although the UPR initially mitigates ER stress and as such confers cytoprotection, persistent and severe ER stress leads to activation of apoptosis that eliminates damaged cells (*Shore et al., 2011*; *Tabas and Ron, 2011*).

By interrogating a large chemical library for small molecules that block PERK signaling, we identified ISRIB as a potent ISR inhibitor, functioning downstream of all eIF2α kinases. ISRIB proves a powerful tool to explore the consequences of acute inhibition of the ISR in cells and animals.

## Results

### Design of cell-based screen for inhibitors of PERK signaling

To identify inhibitors of PERK signaling, we engineered a reporter that allows monitoring of PERK activation in living cells. To this end, we constructed a retroviral vector containing the open-reading

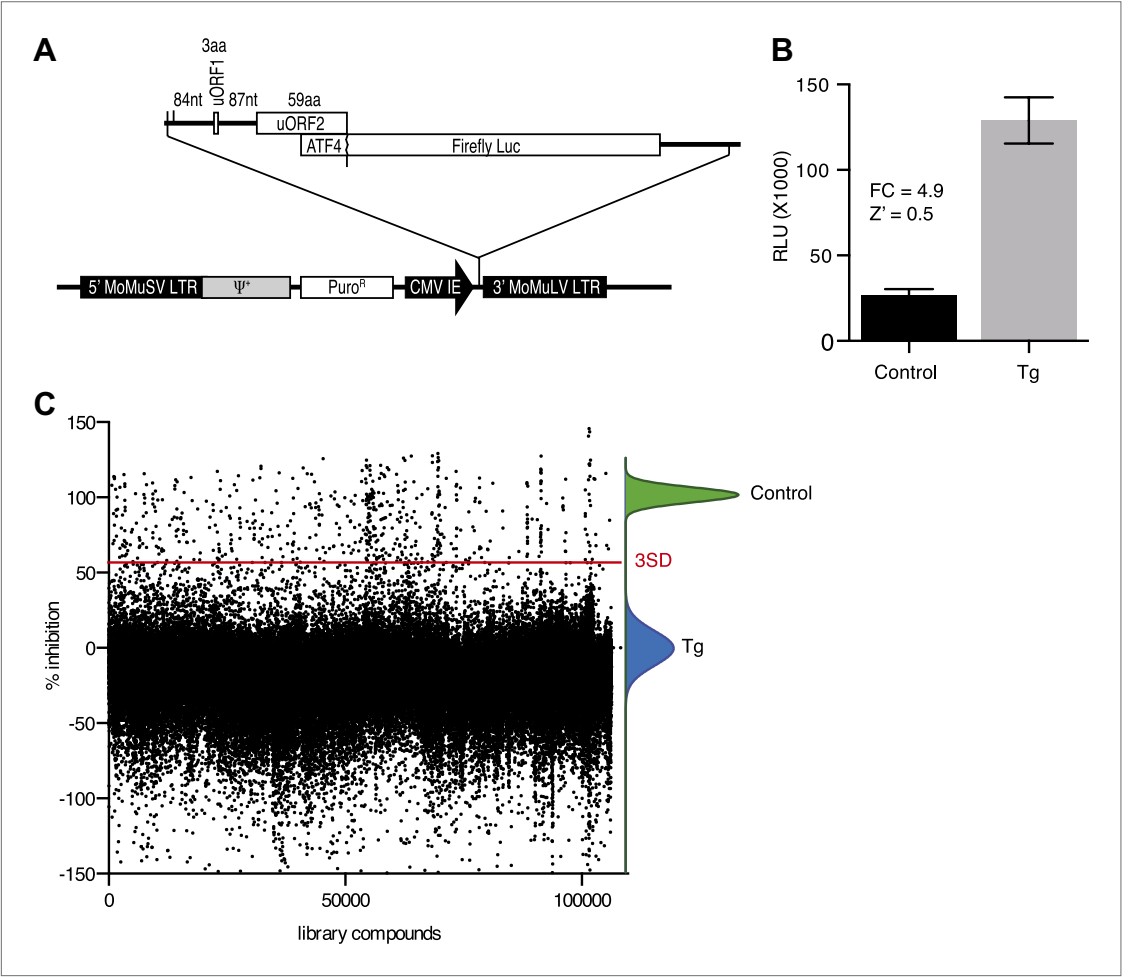

**Figure 1**. High-throughput cell-based screen for inhibitors of PERK signaling. (**A**) Schematic representation of the ATF4 luciferase reporter used in the primary screen. The 5' UTR of human ATF4 containing the uORFs 1 and 2 was fused to firefly luciferase and inserted into a retroviral expression system. (**B**) Primary screen optimization. HEK293T stably expressing the ATF4 luciferase reporter were plated in 384-well plates and treated for 6 hr with 100 nM thapsigargin (Tg) or DMSO as a no ER stress control. Luciferase production was measured at the end point after 6 hr (mean ± SD). The Z' was calculated as 1−(3 [σ Tg + σ DMSO]/[μ Tg−μ DMSO]). (**C**) Primary screen results. The ATF4 luciferase reporter cell line was treated for 6 hr with 100 nM thapsigargin and library compounds (10 μM). Inhibition of the luciferase activity reporter was calculated as the percent reduction in relative luminescence normalized to thapsigargin treatment (0% inhibition) and the no-ER stress control (100% inhibition). Compounds were considered hits if they lied beyond three standard deviations (SD) from the thapsigargin treatment mean (red line).

frame of firefly luciferase fused to the 5′UTR of ATF4 mRNA (*Figure 1A*), which contains two short open-reading frames (uORFs) that control ATF4 translation in a stress-dependent manner. After infection, we established a HEK293T cell line harboring the stably integrated reporter. We used thapsigargin, a potent ER stressor that inhibits the ER calcium pump, to activate PERK and induce eIF2α phosphorylation. Thapsigargin treatment resulted in a 4.9-fold induction in luciferase activity in a 384 well format with a Z factor of 0.5 (*Figure 1B*). This format was used to screen 106,281 compounds covering a wide chemical space. We identified 460 hits (0.43%) (*Figure 1C*), which were further validated in an 8-point dose-response assay using the same reporter. We further triaged the compounds by discarding inhibitors that also affected the IRE1 branch of the UPR using a luciferase-based XBP1 splicing reporter. Less than half (187 hits) of our initial hits proved unique to the PERK branch. We next used an orthogonal secondary screen that employed a different reporter (bi-cistronic ATF4-dGFP-IRES-mCherry) stably integrated into a different cell line (U2OS cells). The read-out of this latter screen was microscopy-based, which allowed us to simultaneously assess acute toxicity by cell counting, further reducing the number of viable hits to 77 (data not shown). As a tertiary screen, we tested compounds for their ability to inhibit ER stress-elicited induction of endogenous ATF4 by Western blot analysis. Twenty-eight compounds passed this test and were analyzed further.

## A symmetric bisglycolamide, ISRIB, is a potent inhibitor of PERK signaling

One of the 28 compounds was of particular interest because of its high potency in cells (library compound $IC_{50}$ = 40 nM). This compound (henceforth referred to as 'ISRIB' for Integrated Stress Response inhibitor) is a symmetric bis-glycolamide, containing a central bi-substituted cyclohexane, and can exist as two diastereomers, *cis* and *trans* (*Figure 2A*). We synthesized both isomers and tested their ability to inhibit the ATF4-luciferase reporter (*Figure 2B*). *Trans*-ISRIB proved 100-fold more potent ($IC_{50}$ = 5 nM) than *cis*-ISRIB ($IC_{50}$ = 600 nM), indicating that the compound's interaction with its cellular target is stereospecific. Given the two-order-of-magnitude difference in activity in this assay, the measured activity of *cis*-ISRIB may be an over-estimate, as we cannot exclude a small contamination with *trans*-ISRIB, which is far more potent. The lower $IC_{50}$ of *trans*-ISRIB relative to the compound in the small molecule library indicates that the library likely contains a mixture of the two stereoisomers. All further experiments in this study were carried out with the synthesized *trans*-isomer of ISRIB.

## ISRIB is PERK-branch specific but does not impair PERK phosphorylation

We next determined at which step ISRIB blocks ATF4 production. To this end, we first probed the phosphorylation status of PERK by Western blotting. PERK phosphorylation is indicative of its activation by autophosphorylation and can be recognized by reduced mobility on SDS-polyacrylamide gels. Notably, ISRIB did not inhibit the mobility shift of PERK observed in ER-stressed cells (*Figure 2C*). Rather, we observed an exaggerated mobility shift, indicative of increased phosphorylation of PERK upon ER stress, induced by either thapsigargin or tunicamycin (an inhibitor of *N*-linked glycosylation). In each case, in the absence of ISRIB, ATF4 and XBP1s were produced upon ER stress induction. In agreement with the behavior of the reporters described above, ISRIB blocked production of endogenous ATF4, whereas XBP1 mRNA splicing (*Figure 2D*) and XBP1s production persisted (*Figure 2C* and *Figure 3—figure supplement 1*). As shown below (*Figure 5D*), ISRIB also did not affect activation of the ATF6-branch of the UPR. We conclude that ISRIB specifically blocks signaling of the PERK-branch of the UPR.

## ISRIB-treated cells are resistant to eIF2α phosphorylation

Given that PERK phosphorylation was not diminished in ISRIB-treated, ER-stressed cells, we next directly assessed eIF2α phosphorylation. We measured the levels of phosphorylated eIF2α using an antiphospho-eIF2α antibody-based assay to quantify phosphorylation at serine 51 (see 'Materials and methods'). Upon induction of ER stress by tunicamycin or thapsigargin, phosphorylation of eIF2α increased over time, reaching a fourfold and sevenfold increase after 120 min respectively (*Figure 3A*). Unexpectedly, ISRIB did not block eIF2α phosphorylation under either of these ER stress-inducing conditions. On the contrary, 120 min after tunicamycin addition, ISRIB further increased the level of eIF2α phosphorylation, approaching that obtained with thapsigargin. ISRIB alone had no effect on eIF2α phosphorylation. These results indicate that ISRIB blocks effects downstream of PERK and eIF2α phosphorylation.

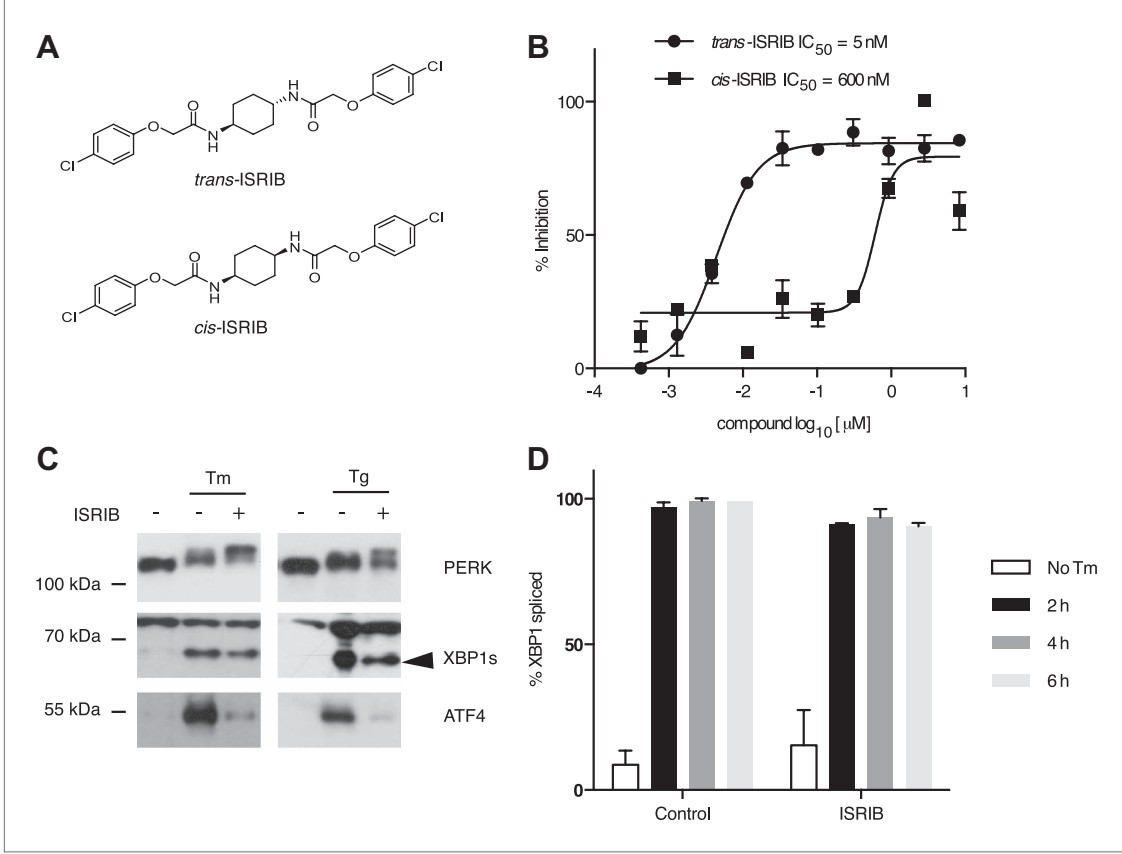

**Figure 2**. Identification of ISRIB as a potent cell-based inhibitor of PERK signaling. (**A**) Structures of ISRIB isosteromers. (**B**) Inhibition of the ATF4 luciferase reporter in HEK293T cells by ISRIB stereoisomers. Inhibition is plotted in relation to the concentration of either the *cis* or *trans* isomer of ISRIB. Cells were treated with 2 µg/ml of tunicamycin to induce ER stress and different concentrations of the inhibitors for 7 hr (N = 2, mean ± SD). (**C**) Effect of ISRIB on production of endogenous ATF4, PERK phosphorylation, and XBP1s production. An immunoblot analysis of PERK, ATF4 and XBP1s in HEK293T cells treated with different ER stress inducers (2.5 µg/ml tunicamycin [Tm] or 100 nM thapsigargin [Tg]) with or without 200 nM ISRIB for 3 hr is shown. The arrowhead marks the XBP1s specific band. (**D**) Effect of ISRIB on XBP1 mRNA splicing. Taqman assays for XBP1unspliced (XBP1u) and XBP1spliced (XBP1s) on cDNA synthesized from total RNA extracted from U2OS cells treated with 2 µg/ml of tunicamycin in the presence or absence of 200 nM ISRIB for the indicated times are shown. Percent splicing was calculated as the ratio of XBP1s over total XBP1 mRNA (XBP1u + XBP1s) (mean ± SD).

One way of explaining why ISRIB blocks ATF4 production yet leaves eIF2α phosphorylation intact is by rendering cells insensitive to the effects of this phosphorylation event. In agreement with this notion, ISRIB sustained global translation (as monitored by $^{35}$S-methionine incorporation into newly synthesized polypeptides) even in the presence of ER stress (*Figure 3B*). After thapsigargin treatment, cells experienced a 40% drop in translation, which was abolished by ISRIB. As predicted by this result, extracts prepared from mouse embryonic fibroblasts (MEFs) experiencing ER stress showed a pronounced increase in the 80S monosomes at the expense of polyribosomes (*Figure 3C*), which was reversed (at least partially) by addition of ISRIB. We chose MEFs for this analysis because they show stronger translational inhibition in response to ER stress than HEK293T cells. ISRIB was the only molecule in our collection of 28 hits that reversed translational attenuation upon ER-stress.

Under normal growth conditions, an abundance of 43S pre-initiation complexes (PICs) leads to mRNAs loaded with a small ribosomal subunit in addition to fully assembled ribosomes. The presence of PICs on an mRNA can be detected as 'halfmer' peaks on polysome gradients. In the gradients shown in *Figure 3C*, addition of a PIC to disomes and trisomes was well resolved (enlarged in *Figure 3—figure supplement 4*). Upon eIF2α phosphorylation in ER-stressed cells, the reduction in PIC resulted in disappearance of the halfmer population. As expected, the disappearance of the halfmer peak upon ER-stress was dependent on eIF2α phosphorylation as no reduction was observed in MEFs that solely express non-phosphorylatable eIF2α(S51A) (*Figure 3—figure supplement 5*). Importantly, ISRIB partially restored the halfmer population in ER-stressed cells, providing support to the notion that it

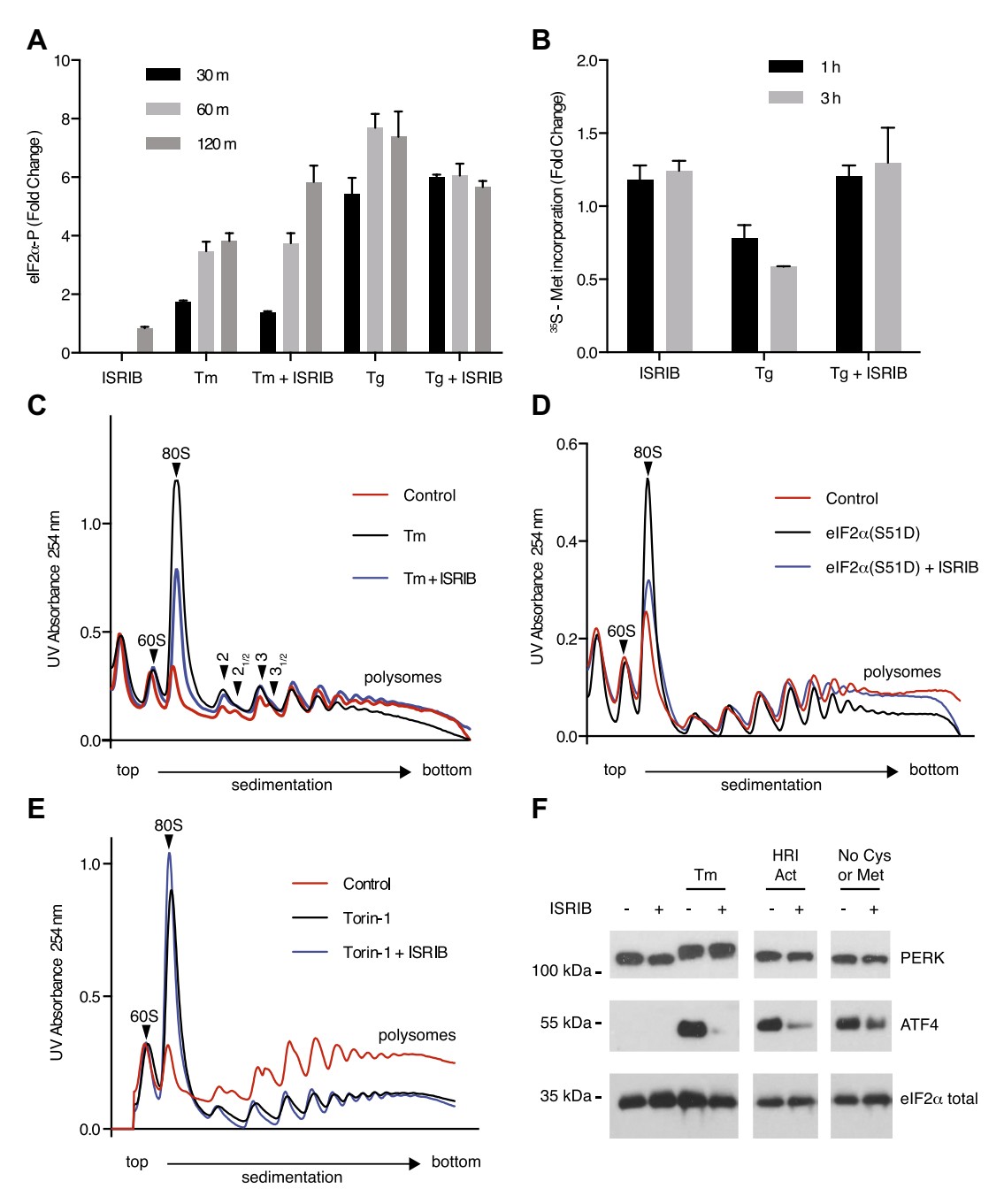

**Figure 3**. ISRIB makes cells resistant to eIF2α phosphorylation. (**A**) ISRIB does not block eIF2α phosphorylation upon ER stress. eIF2α phosphorylation was measured using an alpha-screen Surefire eIF2α p-S51 assay (see 'Materials and methods'). U2OS cells were plated in 96-well plates and treated with 2 µg/ml tunicamycin or 100 nM thapsigargin in the presence or absence of 100 nM ISRIB for the indicated times or with ISRIB alone for 120 m (N = 4, mean ± SD). See *Figure 3—figure supplement 1* for supporting Western blot analysis of eIF2α phosphorylation. (**B**) ISRIB blocks global translational attenuation observed after eIF2α phosphorylation during ER stress. HEK293T cells were treated with 100 nM thapsigargin and 200 nM ISRIB for either 1 or 3 hr prior to a 20 min pulse with 35S methionine before lysis. Equal amounts of lysate were loaded on an SDS-PAGE gel and quantification of radiolabeled methionine incorporation of lysates was done by gel densitometry (N = 2, SD) using ImageJ. (see *Figure 3—figure supplement 2* for SDS-PAGE). (**C**) Polysome gradient analysis showing the block in global translational attenuation upon addition of ISRIB on ER-stressed cells. MEFs were grown in the presence or absence of 2 µg/ml of tunicamycin with or without 200 nM ISRIB for 1 hr. Cell lysates were loaded on a 10–50% sucrose gradient, centrifuged at 150,000×*g* for 2.4 hr and absorbance at 254 nm was measured across the gradient (see *Figure 3—figure supplement 3* for quantitation of polysome profile). A representative experiment is shown (N = 3). See *Figure 3—figure supplement 4* for a close-up of the disome and trisome peaks. (**D**) Cells treated

*Figure 3. Continued on next page*

*Figure 3. Continued*

with ISRIB are resistant to the global translational attenuation exerted by forced expression of eIF2α(S51D). HEK293Trex cells were transduced with a tetracycline inducible phospho-mimetic (S51D) allele of eIF2α. Transgene expression was induced by addition of 25 nM doxycycline for 14 hr in the presence or absence of 200 nM ISRIB. Lysates were collected and analyzed as described in panel (**C**) (see **Figure 3—figure supplement 6** for quantitation of polysome profile). A representative experiment is shown (N = 2). (**E**) ISRIB does not reverse global translational attenuation exerted through inhibition of CAP-dependent initiation. Wild-type MEFs were treated with 750 nM Torin-1 in the presence or absence of 200 nM ISRIB for 2 hr. Lysates were collected and analyzed as described in panel (**C**). A representative experiment is shown (N = 2). (**F**) ISRIB blocks production of ATF4 upon GCN2 or HRI activation. An immunoblot analysis of PERK, ATF4 and total eIF2α in HEK293T cells starved for cysteine and methionine or treated with an HRI activator (6 μM) for 5 hr in the presence or absence of 200 nM ISRIB is shown. Tunicamycin was used as a positive control for induction of ATF4 and the shift in PERK mobility. Under amino acid starvation we consistently observe a partial reduction of ATF4 production by ISRIB by Western blot analysis but observe a complete block in induction of the ATF4 luciferase reporter (see **Figure 3—figure supplement 7**).

The following figure supplements are available for figure 3:

**Figure supplement 1**. ISRIB does not inhibit eIF2α phosphorylation or XBP1s production.

**Figure supplement 2**. ISRIB blocks translational attenuation upon ER stress.

**Figure supplement 3**. ISRIB blocks translational attenuation upon ER stress.

**Figure supplement 4**. ISRIB partially restores the halfmer population in ER stressed cells.

**Figure supplement 5**. Disappearance of the halfmer peaks upon ER-stress is dependent on eIF2α phosphorylation.

**Figure supplement 6**. ISRIB sustains translation upon expression of eIF2α(S51D).

**Figure supplement 7**. ISRIB blocks induction of the ATF4 luciferase translational reporter upon HRI and GCN2 activation.

helps maintain high PIC levels even when eIF2α is phosphorylated (*Figure 3—figure supplement 4*). These data indicate that ISRIB exerts its function by maintaining elevated ternary complex levels.

To further ascertain that cells treated with ISRIB are resistant to the effects of eIF2α phosphorylation, we transduced an inducible phospho-mimetic allele of eIF2α in which serine 51 was changed to an aspartic acid (S51D) into HEK293T cells. Expression of this allele upon doxycycline addition induced translational attenuation (*Figure 3D*) as seen by an increase in the 80S peak and a decrease in the polysome population. ISRIB rescued translation returning it to the levels observed in non-induced cells. In conclusion, ISRIB restores translation in cells containing either phospho-eIF2α or eIF2α(S51D), thereby excluding any pleiotropic effects that might have been caused by the reagents used to activate ER stress.

To rule out that ISRIB exerts non-specific effects on translation independent of eIF2α phosphorylation, we tested whether ISRIB reverses a translational block in CAP-mediated initiation. To this end we used Torin-1, an inhibitor of mTOR that blocks phosphorylation of 4E-BP1 and S6K1, and leads to translational attenuation (*Thoreen et al., 2012*). Addition of Torin-1 to MEFs led to an increase in the 80S peak and reduction in the polysome population to a similar extent as shown above in cells treated with ER stressors or expressing eIF2α(S51D) (*Figure 3E*, compare with *Figure 3C,D*). In contrast to these treatments, addition of ISRIB did not reverse the effect of Torin-1 on translation. Therefore, the ability of ISRIB to block translational attenuation is specific to eIF2α phosphorylation.

If ISRIB makes cells insensitive to eIF2α phosphorylation, it should not matter which kinase phosphorylates eIF2α. To test this prediction, we subjected cells to amino acid starvation, which activates the eIF2α kinase GCN2 and leads to ATF4 production. In addition, we used a recently identified small molecule activator to induce eIF2α phosphorylation by activating HRI, another eIF2α kinase (*Chen et al., 2011*). As expected, ISRIB blocked ATF4 induction after activation of either GCN2 or HRI (*Figure 3F*). Under both conditions, PERK was not activated as shown by a lack of mobility shift. These data suggest that ISRIB is a bona fide ISR inhibitor that blocks signaling downstream of all eIF2α kinases.

Both *DDIT3* (the gene encoding CHOP) and *PPP1R15A* (the gene encoding GADD34) are transcriptional targets of ATF4. Thus, blocking ATF4 accumulation with ISRIB should result in a reduction in the transcriptional induction of the mRNAs encoding these targets. As shown in *Figure 4A*, GADD34 and CHOP mRNAs accumulated in ER-stressed U2OS cells, and ISRIB significantly reduced their induction.

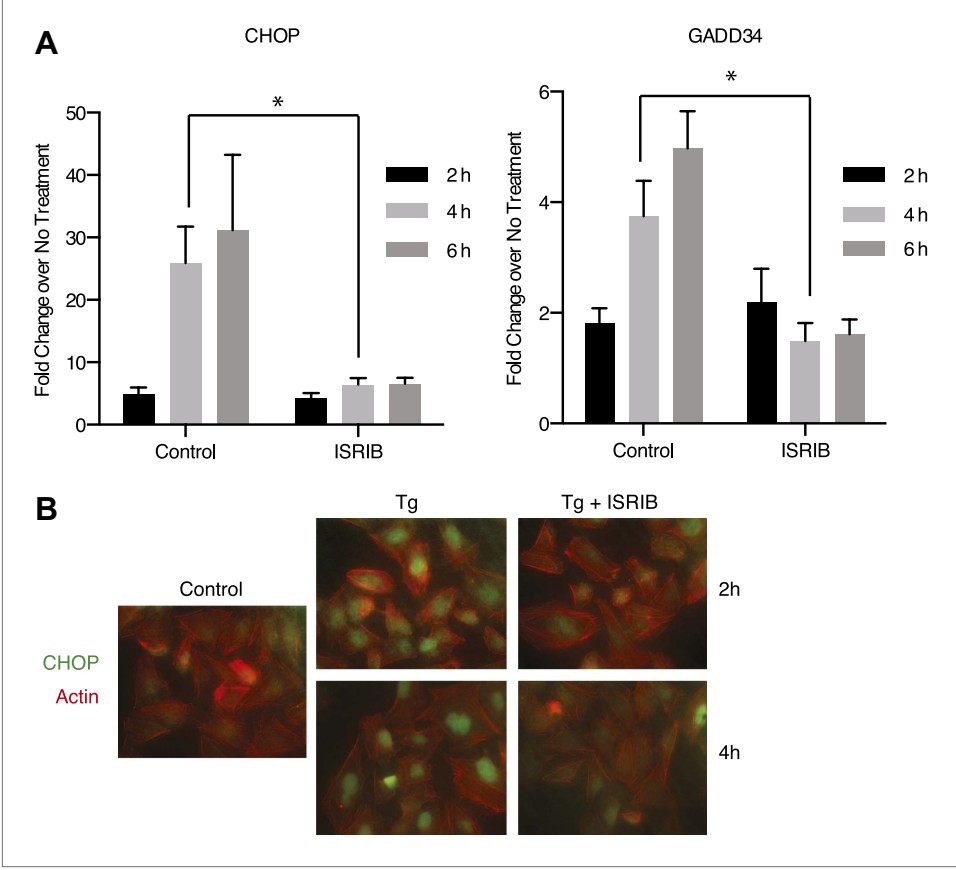

**Figure 4**. ISRIB impairs induction of the transcriptional network controlled by ATF4. (**A**) ER-Stress dependent induction of CHOP and GADD34 mRNA is impaired in cells treated with ISRIB. qPCR analysis of total RNA extracted from U2OS cells treated with 2 µg/ml of tunicamycin in the presence or absence of 200 nM ISRIB for the indicated times. mRNA levels for each sample were normalized to GAPDH (N = 4, mean ± SD). p values are derived from a one-tail Student's t-test for unpaired samples. Statistical significance: CHOP, *p=0.0006; GADD34, *p=0.0008. (**B**) ISRIB blocks CHOP production during ER stress. An immunofluorescence analysis of U2OS cells treated with 100 nM thapsigargin for 2 or 4 hr in the presence or absence of 200 nM ISRIB is shown. A secondary Alexa Dye 488 anti-mouse antibody and rhodamine-phalloidin were used to visualize CHOP and actin, respectively.

In agreement, we observed no CHOP accumulation after induction of ER stress in ISRIB-treated cells (*Figure 4B*). Thus ISRIB impairs the transcriptional network governed by ATF4 during the ISR.

## ISRIB impairs adaptation to ER stress

As previously shown, cells homozygous for non-phosphorylatable eIF2α, eIF2α(S51A), are unable to cope with ER stress properly, leading to reduced viability (*Lu et al., 2004*). This indicates that events downstream of eIF2α phosphorylation are required to resolve the stress. As shown in *Figure 5A*, ISRIB treatment of wild-type cells had similar consequences. Importantly, addition of ISRIB alone did not affect cell viability, as judged by the number of colonies that form after acute treatment. By contrast, ISRIB addition caused a strong synergistic effect on ER-stressed cells, reducing colony number and size significantly more than ER-stress alone. This reduction in cell survival resulted from activation of apoptosis as the activity of the executioner caspases 3 and/or 7 was significantly induced under these conditions (*Figure 5B*; *Salvesen and Ashkenazi, 2011*).

The notion that ER stress remains unmitigated in ISRIB-treated cells is supported by sustained activation of all three UPR sensors. First, as shown in *Figure 2C*, PERK was hyper-phosphorylated. Second, cells expressing an IRE1-GFP fusion protein showed prolonged foci formation (*Figure 5C*), indicative of IRE1 oligomerization. Third, we observed prolonged ER stress-induced proteolytic

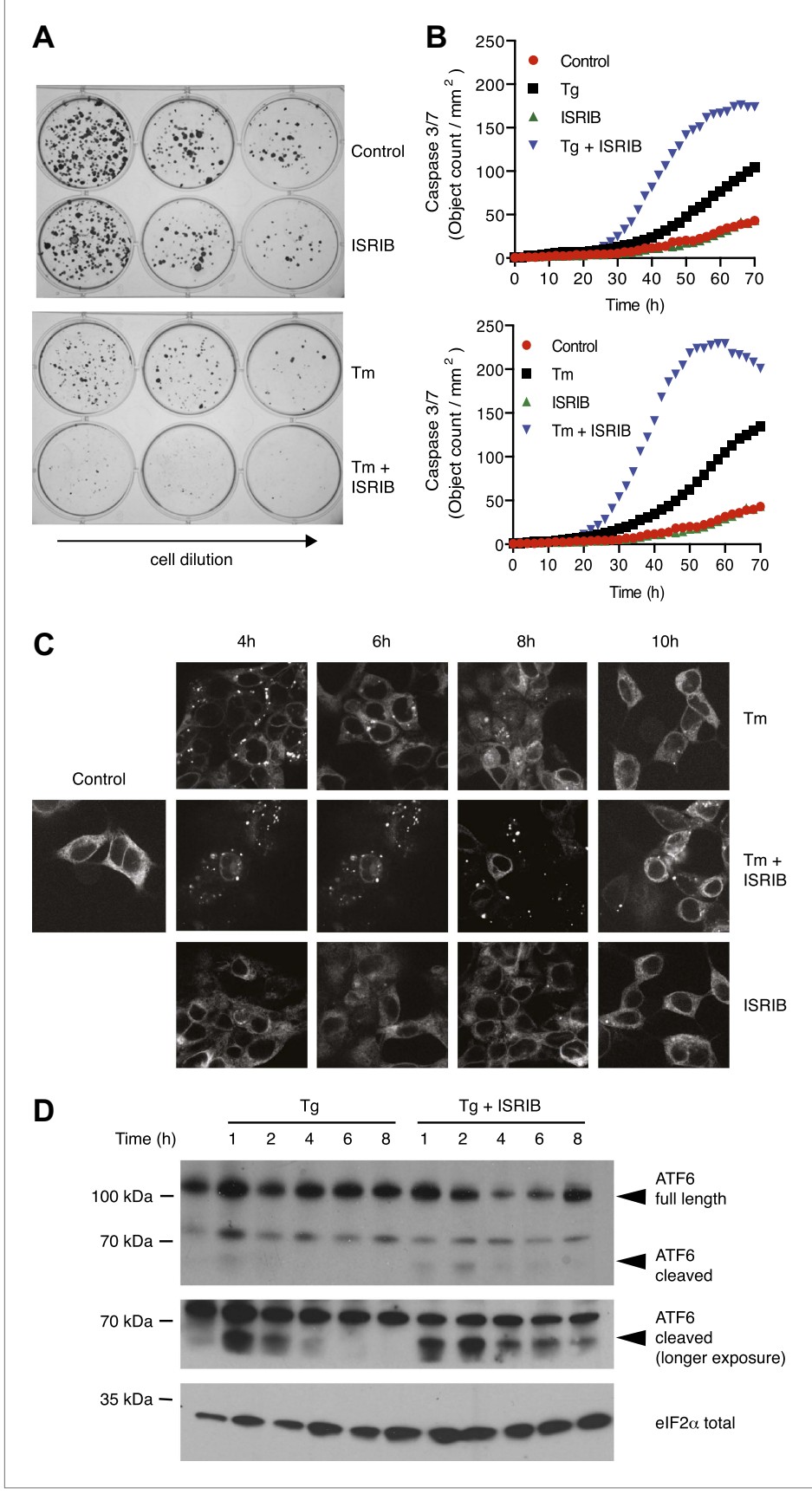

Figure 5. Continued on next page

*Figure 5. Continued*

**Figure 5**. ISRIB impairs adaptation to ER-stress prolonging activation of the UPR sensors. (**A**) ISRIB sensitizes cells to acute ER stress. HEK293T cells were subjected with an acute dose of tunicamycin (2 μg/ml), ISRIB (200 nM) or a combination of both for 24 hr. The treated cells were equally diluted to a concentration that would allow single cell clonal expansion and re-seeded onto six-well plates in a threefold dilution series. Clonal colonies were visualized by Crystal Violet stain. (**B**) ISRIB synergizes with ER stress to activate caspase 3/7. Hela cells were plated in 96-well plates and treated with 5 μg/ml of tunicamycin or 500 nM thapsigargin with or without 25 nM ISRIB for the indicated times. Caspase3/7 activation was measured using Cellplayer kinetic caspase 3/7 reagent and cells were imaged in an IncuCyte system. Green object count/mm$^2$ representing caspace-3/7 activation was measured at 2 hr intervals (See **Figure 5—figure supplement 1** for endpoint quantitation of % cells with activated caspase 3/7). (**C**) IRE1 oligomers are sustained on ER-stressed cells treated with ISRIB. Confocal microscopy micrographs of HEK293Trex cells carrying an inducible GFP-tagged IRE1 allele were treated with 10 nM doxycycline for 24 hr to induce the transgene, followed by treatment with 5 μg/ml of tunicamycin in the presence or absence of 200 nM ISRIB for the indicated times. (See **Figure 5—figure supplement 2** for corresponding XBP1 mRNA splicing data). (**D**) ATF6 cleavage is sustained in ER-stressed cells treated with ISRIB. Immunoblot analysis of ATF6 processing in HEK293Trex cells carrying an inducible FLAG epitope-tagged ATF6. Cells were treated with 50 nM doxycycline for 18 hr to induce the transgene followed by treatment with 100 nM thapsigargin in the presence or absence of 200 nM ISRIB for the indicated times. Total eIF2α is used as a loading control.

The following figure supplements are available for figure 5:

**Figure supplement 1**. ISRIB synergizes with ER-stress to induce caspase 3/7.

**Figure supplement 2**. XBP1 splicing is sustained in ER-stressed cells upon addition of ISRIB.

---

processing of ATF6 (*Figure 5D*). Importantly, in the absence of ER stress ISRIB treatment alone did not induce any of these sensors (*Figure 3—figure supplement 1*; *Figure 5C* and data not shown).

## ISRIB increases long-term memory in rodents

eIF2α$^{+/S51A}$ (Eif2s1$^{+/S51A}$) heterozygote mice display enhanced memory, while induction of the eIF2α kinase PKR in brain pyramidal cells impairs memory (*Costa-Mattioli et al., 2007*; *Jiang et al., 2010*). Based on these observations, we wondered whether treatment of mice with ISRIB would affect memory. ISRIB showed favorable properties in pharmacokinetic profiling experiments indicating sufficient bioavailability for in vivo studies. ISRIB displayed a half-life in plasma of 8 hr (*Figure 6A*) and readily crossed the blood-brain barrier, quickly equilibrating with the central nervous system (*Figure 6B*). After a single intraperitoneal injection, we detected ISRIB in the brain of mice at concentrations several fold higher than its IC$_{50}$ (24 hr after injection, the ISRIB concentration in the brain was approximately 60 nM). To explore ISRIB's effects on memory, we injected mice intraperitoneally with ISRIB and tested hippocampus-dependent spatial learning. To this end, we trained mice in a Morris water maze, in which animals learn to associate visual cues with the location of a submerged hidden platform. Because we were looking for memory enhancement, we used a weak training protocol. As shown in *Figure 6C*, ISRIB-treated mice reached the hidden platform significantly faster (escape latency after 5 days of training = 16.4 ± 4.8 s) compared to vehicle treated controls (68.1 ± 20 s, p<0.05). The difference was already pronounced by days 3 and 4. In agreement with these results, ISRIB-treated mice significantly preferred the target quadrant in a 'probe test' conducted at the end of the training sessions, in which the platform was removed from the pool (p<0.05; *Figure 6D*) and showed increased crossing of the platform location (p<0.05; *Figure 6E*).

We next tested contextual fear conditioning, which represents a different kind of hippocampus-dependent learning in which eIF2α phosphorylation has also been implicated to play a role (*Costa-Mattioli et al., 2007*). In these experiments, we paired a particular environmental context (a different cage) with a foot shock. In this case the context acts as the 'conditioned stimulus, CS' and is associated with the foot shock, the 'unconditioned stimulus, US'. ISRIB-treated mice showed increased freezing upon presentation of the conditioned environment 24 hr after training as compared to vehicle treated mice (p<0.05; *Figure 6F*). No differences were observed in short-term memory (1 hr) between these two treatments. Taken together, we conclude that treatment with ISRIB enhances both hippocampus-dependent spatial learning and hippocampus-dependent contextual fear conditioning.

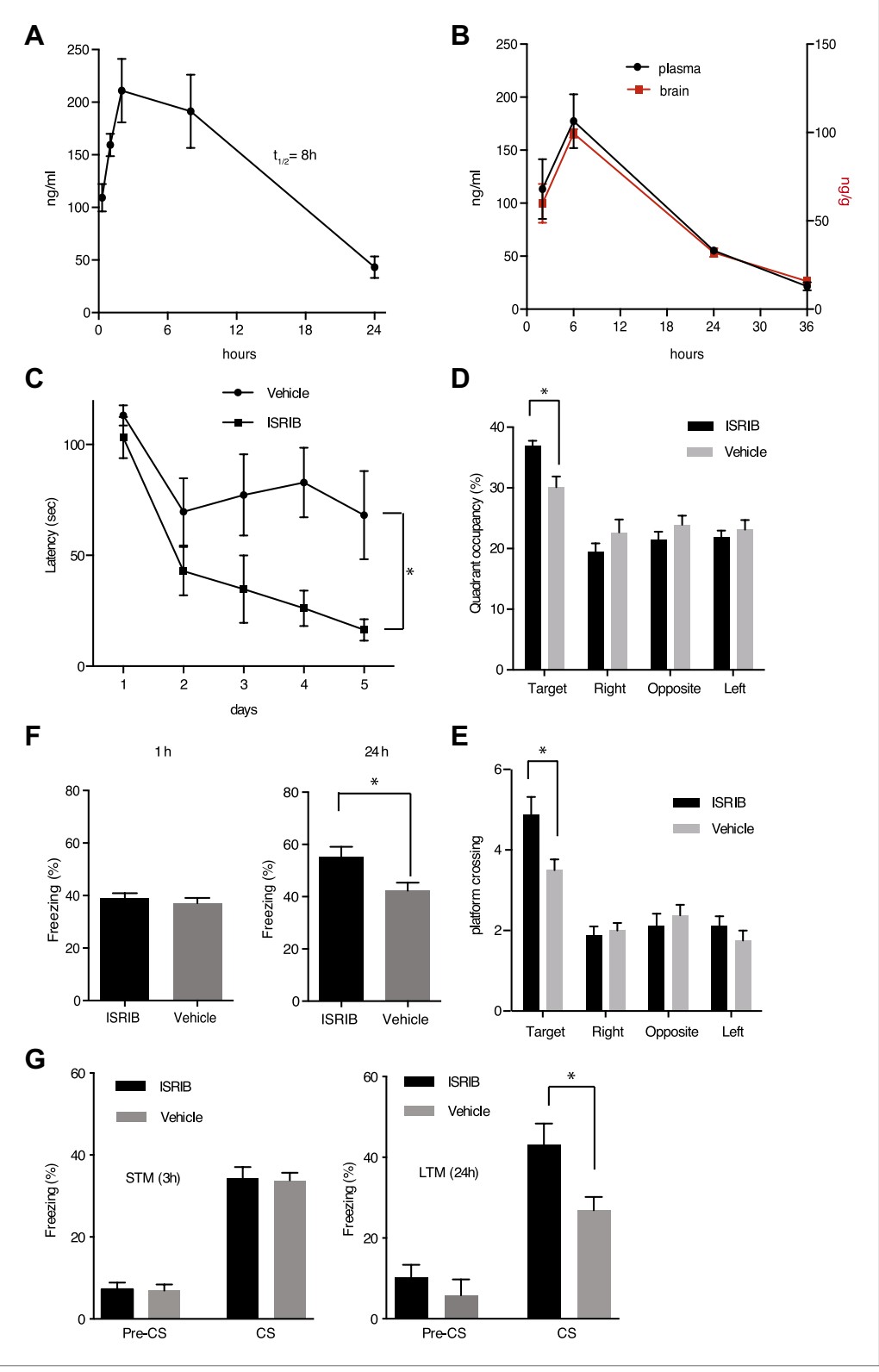

**Figure 6**. ISRIB enhances spatial and fear-associated learning in rodents. (**A**) Plasma concentration (ng/ml) of ISRIB after a single intraperitoneal injection (5 mg/kg). Plasma was collected at the indicated times and the concentration was determined by mass spectrometry (mean ± SEM, N = 3). (**B**) Brain (ng/g tissue) and plasma concentrations (ng/ml) of ISRIB after a single intraperitoneal injection (2.5 mg/kg). Data (mean ± SEM, N = 3) were obtained at the

*Figure 6. Continued on next page*

*Figure 6. Continued*

indicated times. (**C**) Escape latencies are significantly shorter in mice treated with ISRIB. Data (mean ± SEM) were obtained in a weak 5 days-long training session in the hidden platform version of the Morris water maze (1 trial per day). Mean escape latencies were plotted as a function of training days in mice treated with ISRIB (closed squares, N = 8) or vehicle (open circles N = 8) (*p<0.05). Mice were injected daily with ISRIB immediately after training. (**D**) After completion of training in the study shown in panel (**A**), mice treated with ISRIB (black column) showed a significant preference for the target quadrant (*p<0.05). The probe test was performed 24 hr after the last training session. p values are derived from a two-tailed Student's t test for unpaired samples. (**E**) After completion of training in the study shown in panel a, mice treated with ISRIB (black column) increased the number of times they crossed the platform location as compared to the vehicle-treated mice (grey column) (*p<0.05). p values are derived from a two-tailed Student's t test for unpaired samples. (**F**) Systemic administration of ISRIB (intraperitoneally after training) enhances long-term contextual fear memory (right bars, 24 hr), while it does not affect short-term memory (left bars, 1 hr) (N = 10 per group, *p<0.05). Data are presented as mean ± SEM. (**G**) Auditory fear conditioning is enhanced in rats treated with ISRIB. Freezing in response to a tone was assessed 3 hr (short-term memory, STM, left panel) and 24 hr (long-term memory, LTM, right panel) after training (vehicle-treated N = 8, and ISRIB-treated N = 7) after tone presentation (CS) and before tone presentation (pre-CS). For these experiments vehicle or ISRIB was infused directly by cannula into the amygdala after training. ISRIB-infused rats show increase freezing at 24 hr (*p<0.05).

To test learning associated with a different region of the brain, we explored the effects of ISRIB on auditory fear conditioning, which depends on the amygdala. In this type of learning a tone (CS) is paired with a foot shock (US). In these experiments, we injected ISRIB or vehicle directly into the amygdala of rats via cannulation. ISRIB-treated rats showed a significant increase over vehicle-injected rats in the level of freezing when presented with the tone (CS) at 24 hr (long-term memory, p<0.05; *Figure 6G*). By contrast, we observed no difference between ISRIB- and vehicle-treated rats at 3 hr (short-term memory). As expected, both ISRIB- and vehicle-treated rats showed similar freezing responses prior to training (pre-CS). Taken together, these data suggest that long-term memory is selectively enhanced in ISRIB-treated animals.

## Discussion

We identified a novel small molecule, ISRIB, that renders cells resistant to the effects of eIF2α phosphorylation, restoring the cell's translation capacity. ISRIB is the first reported antagonist of the ISR. It acts as a potent and stereospecific inhibitor with an $IC_{50}$ of 5 nM in cultured cells, suggesting a specific and tight interaction with its cellular target. By blocking signaling through the PERK branch of the UPR, ISRIB prevents cells from re-establishing ER homeostasis. Unmitigated ER stress synergizes with ISRIB to induce apoptosis. ISRIB shows good pharmacokinetic properties, readily crosses the blood-brain barrier, and exhibits no overt toxicity in mice, making it suitable for in vivo studies. As such, ISRIB emerges as a powerful tool to explore the roles of the UPR and the ISR in disease models and physiological processes. In particular, we utilized ISRIB to show that overriding the consequences of eIF2α phosphorylation enhances memory consolidation in rodents, suggesting an important role of eIF2α phosphorylation in modulating higher-order brain function.

### Molecular action of ISRIB

The structure of ISRIB provides no insights as to its target in cells. To date, we have synthesized and assayed more than 75 analogs, which demonstrate a tractable structure-activity relationship (to be published elsewhere). The analyses have identified sites on the molecule where affinity tags and/or crosslinking moieties can be added, which promise to aid in target identification. Based on previous insights on how cells can become resistant to eIF2α phosphorylation, we consider two likely scenarios by which ISRIB could act:

- First, ISRIB could weaken the effects of the non-productive interaction of phospho-eIF2α with eIF2B, thereby increasing the available eIF2α-GEF activity in the cell, restoring the concentration of ternary complex that can engage in translation initiation. Precedence for this possibility derives from genetic studies in *Saccharomyces cerevisiae*, where the molecular mechanism of regulation by eIF2α phosphorylation was first discovered. As in mammalian cells, amino acid starvation in yeast leads to GCN2 activation and eIF2α phosphorylation, resulting in overall translational down-regulation

and translational induction of a transcriptional activator, GCN4, mediated by uORFs in the 5'UTR of its mRNA (*Hinnebusch, 2005*). eIF2B is a conserved protein complex comprised of five different subunits, two of which form the catalytic core, and the remaining three have regulatory roles. Mutations in different eIF2B subunits can elicit a phospho-eIF2α resistant phenotype (*Vazquez de Aldana and Hinnebusch, 1994*; *Pavitt et al., 1997, 1998*). These mutations have been proposed to act either by weakening the interaction of phospho-eIF2α with eIF2B, reducing its ability to outcompete non-phosphorylated eIF2 for binding, or by allowing binding of phospho-eIF2α to the mutant eIF2B in a manner that is conducive to nucleotide exchange. ISRIB could be altering the affinity of phospho-eIF2α for eIF2B or overcoming the nonproductive interaction that blocks GTP loading, mimicking the effect of these mutations.

- Second, ISRIB could increase the activity of eIF2B, so that the residual amount not engaged with phospho-eIF2α is sufficient to sustain normal levels of ternary complex. Precedence for this possibility derives from studies in macrophages, where engagement of toll-like receptor (TLR) 4 results in activation of the catalytic activity of eIF2B (*Woo et al., 2012*). This activation results from engagement of the TLR-signaling pathway that induces a phosphatase removing a constitutively present inhibitory phosphate from the eIF2B ε-subunit (S539). Pathogens utilize this mechanism to circumvent translational attenuation and CHOP production under prolonged stress-inducing conditions (*Woo et al., 2009*). ISRIB did not reduce phosphorylation of S539 in the eIF2B ε-subunit (*Figure 3—figure supplement 1*), indicating that it does not utilize this particular regulatory phosphorylation to increase GEF activity. However, ISRIB may modulate other post-translational modifications that impinge on the activity of eIF2B.

## ISRIB can influence cell fate

As a signaling network with interconnected signaling branches, the UPR exhibits both cytoprotective and pro-apoptotic functions. When faced with ER stress, PERK-mediated translational attenuation contributes to adaptation by reducing the load of newly synthesized proteins that are translocated into the ER (*Harding et al., 2000*). In addition, induction of the transcription regulator ATF4 upregulates many genes that increase the protein folding capacity in the ER. Both of these activities serve to reestablish homeostasis, balancing the protein folding load and protein folding capacity in the ER lumen. This reasoning is supported by the increased sensitivity to ER stress exhibited by MEFs that lack PERK or ATF4, as well as MEFs that carry a non-phosphorylatable knock-in allele of eIF2α(S51A) (*Harding et al., 2000*; *Harding et al., 2003*; *Lu et al., 2004*). In agreement, we show that ISRIB decreases the viability of cells that are subjected to ER-stress. In these cells, ISRIB sustains IRE1 and ATF6 activation, indicating that ER stress remains unmitigated in the absence of PERK signaling. As some cancer cells sustain an activated UPR to aid in their survival, ISRIB could provide a new therapeutic approach to cancer chemotherapy. In agreement, a PERK-specific inhibitor demonstrates antitumor activity in a human pancreatic tumor xenograft model (*Axten et al., 2012*). The deleterious synergistic effect between ER-stress and ISRIB may be generally advantageous to kill cancer cells, especially those derived from secretory lineages that have increased secretory load and increased basal levels of ER stress (including myelomas, and pancreatic and breast cancers).

Importantly, by acting downstream of eIF2α phosphorylation, ISRIB blocks multiple stress effectors (i.e., all eIF2α kinases). During tumor growth, hypoxic conditions and a lack of nutrients can activate both PERK and GCN2, and PERK$^{-/-}$ or GCN2$^{-/-}$ MEFs give rise to significantly smaller tumors in mouse xenograft models than their wild-type counterparts (*Bi et al., 2005*; *Ye et al., 2010*). Hence both kinases have pro-survival roles in tumor development. By blocking signaling by both kinases, ISRIB displays unique properties that may be beneficial in reducing cellular fitness of tumor cells.

## ISRIB and brain function

The importance of eIF2/eIF2B function in the human brain is underscored by familial diseases caused by mutations in these factors. One example is Childhood Ataxia with CNS Hypomyelination (CACH), also known as Vanishing White Matter disease (VWM), which has been mapped to mutations in different subunits of eIF2B (*Li and Wang, 2004*). A second example links a familial intellectual disability syndrome to a mutation in the γ-subunit of eIF2 complex (*Borck et al., 2012*).

Several lines of genetic evidence in mice suggest that phosphorylation-dependent regulation of eIF2α phosphorylation is a critical hub for the control of synaptic plasticity (as assessed by late long-term potentiation [L-LTP] in brain slices) and memory consolidation (as assessed in behavioral tasks in

animals). In particular, the threshold for induction of L-LTP is reduced and memory consolidation is enhanced in mice lacking GCN2 or PKR and in mice heterozygous for non-phosphorylatable eIF2α(S51A), which have reduced levels of eIF2α phosphorylation (*Costa-Mattioli et al., 2005*; *Costa-Mattioli et al., 2007*; *Zhu et al., 2011*). As we show here, ISRIB pharmacologically phenocopies these genetic manipulations in behavioral tasks by rendering cells insensitive to eIF2α phosphorylation. In agreement, treatment of mice with a PKR inhibitor was reported to enhance memory consolidation, and, conversely, treatment with salubrinal, an inhibitor that prolongs eIF2α phosphorylation, to block L-LTP and memory consolidation (*Costa-Mattioli et al., 2007*; *Zhu et al., 2011*).

eIF2α phosphorylation results in a dual effect on gene expression: a global translational diminution and translational upregulation of select mRNA, including ATF4 mRNA. Both may be important to explain the observed effects on L-LTP and memory. It has long been appreciated that new protein synthesis is required for memory consolidation and that ATF4 represses CREB-mediated transcription of 'memory genes' (*Klann and Dever, 2004*; *Sutton and Schuman, 2006*). Indeed, this latter function of ATF4 in memory consolidation is evolutionarily conserved from *Aplysia* to rodents (*Yin et al., 1994*; *Bartsch et al., 1995*; *Chen et al., 2003*). Because a small physiological increase in the level of eIF2α phosphorylation that does not significantly alter overall translation is sufficient to induce ATF4, production of this transcription factor can be finely tuned in neuronal cells by perhaps selective activation of different eIF2α kinases. The observed effects of ISRIB may therefore result from overcoming effects caused by a relatively small level of regulatory phosphorylation that is distinct from the high level resulting from ER stress-inducing agents. In light of this reasoning, a therapeutic window may exist in which ISRIB's effects as memory enhancer can be exploited without encountering long-term toxic consequences.

ISRIB increases memory consolidation, allowing pharmacological enhancement of the brain's ability to learn. Evolution therefore did not arrive at a maximally optimized process, imposing a brake (via eIF2α phosphorylation) on memory consolidation. This mechanism may underscore the importance of filtering memories before committing them to long-term storage. Indeed, eIF2α phosphorylation also plays a role in dynamic restructuring of memory, as indicated by studies showing that ablation of PERK in the brain impairs behavioral flexibility (*Trinh et al., 2012*). Our findings raise the possibility that ISRIB or compounds with related activities could serve as invaluable tools in deciphering these higher order brain functions and perhaps be beneficial as a therapeutic agent effecting memory improvement in diseases associated with memory impairment.

## Materials and methods

### Cell culture
HEK293T, TREx293, U2OS, Hela, and mouse embryonic fibroblasts (MEFs) were maintained at 37°C, 5% $CO_2$ in DMEM media supplemented with 10% FBS, L-glutamine and antibiotics (penicillin and streptomycin).

### Chemicals
Tunicamycin was obtained from Calbiochem EMB Bioscience, Billerica, MA. Thapsigargin was obtained from Sigma-Aldrich, St Louis, MO. Torin-1 was obtained from Tocris, MN. The HRI activator was purchased from Maybridge (KM09748), Cornwall, UK.

### Generation of ATF4 reporter constructs and cell lines for small-molecule screening
ATF4 reporters were constructed by fusing the human full-length ATF4 5'-UTR (NCBI Accession BC022088.2) in front of the firefly luciferase (FLuc) or a destabilized eGFP (dEGFP) coding sequences lacking the initiator methionine.

The ATF4-FLuc reporter was generated by cloning a PCR-product containing the ATF4 full-length 5'-UTR (from +1 position a the transcription start site down to one nucleotide after the terminator codon of the second uORF) flanked by KpnI/XhoI and BglII sites at the 5' and 3' ends, respectively, into the KpnI-BglII sites of pCAX-F-XBP1-Luc (kind gift of Takao Iwawaki, RIKEN, Hirosawa, Japan). The resulting construct, pCAX-ATF4-FLuc, was then digested with BamHI, blunted with T4 DNA polymerase, and then digested with XhoI. The resulting fragment was then subcloned into the retroviral expression vector pLPCX (Clontech, Mountain View, CA) after digesting it with HindIII, blunting with T4 DNA polymerase and then digesting with XhoI to generate pLPCX-ATF4-FLuc (DAA-312). DAA-312 was

used to produce recombinant retroviruses using standard methods and the resulting viral supernatant was used to transduce HEK293T cells, which were then subsequently selected with puromycin to generate a stable cell line employed in the primary screen.

The ATF4-dEGFP reporter was generated using a PCR fusion-based approach. A PCR product containing the ATF4 full-length 5' leader sequence (from +1 position a the transcription start site) fused to the eGFP coding sequence 1 nucleotide downstream of the terminator codon of the second uORF, and flanked by BamHI and EcoRI, was cloned into the cognate sites of pEGFP-N3 (Clontech, Mountain View, CA) to generate pCMV-ATF4-eGFP. To destabilize the eGFP fusion protein and increase the dynamic range of the reporter, residues 422–461 of mouse ornithine decarboxylase (mODC1), corresponding to its PEST sequence (*Li et al., 1998*), were fused to the C-terminus of the ATF-eGFP fusion protein. To such end, the corresponding mODC1 coding sequence was amplified by PCR and cloned into the BstXI and EcoRI sites of pCMV-ATF4-eGFP. The resulting construct was designated pCMV-ATF4-d2EGFP. To further destabilize the ATF4-d1EGFP fusion protein, alanine substitutions E428A, E430A, E431A (*Li et al., 1998*) were introduced in the ODC1 PEST sequence to generate pCMV-ATF4-d1EGFP. The ATF4-d1EGFP coding sequence was then excised from the expression vector using BamHI and EcoRI and subcloned into the BglII-EcoRI sites of the retroviral expression vector pLPCX (Clontech, Mountain View, CA) to generate pLPCX-ATF4-d2EGFP. Lastly, a fusion PCR product containing the encephalomyocarditis virus internal ribosomal entry site (EMCV-IRES) upstream of the monomeric cherry (mCherry) coding sequence and flanked by EcoRI and NotI recognition sites was subcloned into the cognate sites of pLPCX-ATF4-d1EGFP, thereby generating pLPCX-ATF4-d1EGFP-IRES-mCherry (DAA-361). DAA-361 was used to produce recombinant retroviruses using standard methods and the resulting viral supernatant was used to transduce U2OS cells, which were then subsequently selected with puromycin to generate a stable cell line employed in the secondary screen.

## Generation of the inducible eIF2α phosphomimetic mutant construct and cell line

The coding sequences of wild-type mouse eIF2α, phosphomimetic (S51D) mutant was amplified by PCR from a mammalian expression vector (kind gift of David Ron). BamHI and EcoRI recognition sites were engineered into the primers. In addition a Kozak consensus sequence and a N-terminal FLAG epitope tag were engineered in the forward primer. The resulting PCR products were subcloned into the cognate sites of the tetracycline-inducible retroviral expression vector pRetroX-Tight-Pur-GOI (Clontech, Mountain View, CA). 293T target cells stably expressing the reverse tetracycline transactivator (rtTA) were generated by standard retroviral transduction using VSV-G pseudotyped retroviruses encoding rtTA (pRetroX-Tet-On Advanced; Clontech, Mountain View, CA) and selected with G418. These cells were subsequently transduced with a VSV-G pseudotyped retrovirus, encoding the eIF2α(S51D) (DAA-A681) mutant allele, resulting in a puromycin-selected, tetracycline inducible, stable cell line.

## Generation of the inducible 6xHIS-3xFLAG-hsATF6-alpha cell line

6xHis-3xFLAG-hsATF6-alpha was generated by PCR from p3xFLAGCMV7.1-ATF6 (*Shen et al., 2002*) and cloned into pcDNA5/FRT/TO. pcDNA5/FRT/TO-6xHis-3xFLAG-hsATF6-alpha was co-transfected with pOG44 into Flp-In TRex cells (*Cohen and Panning, 2007*) according to manufacturers instructions (Invitrogen, Carlsbad, CA). After selection with 100 µg/ml Hygromycin B (Gold Biotechnology, St Louis, MO) single colonies were isolated, expanded and tested for expression of tagged ATF6.

## High-throughput primary screen

HEK293T cells carrying the ATF4 luciferase reporter were plated on poly-lysine coated 384-well plates (Greiner Bio-one, Monroe, NC) at 30,000 cells per well. Cells were treated the next day with 100 nM thapsigargin and 10 µM of the library compounds (diversity library of 106,281 compounds) for 6 hr. Luminescence was measured using One Glo (Promega, Madison, WI) as specified by the manufacturer. The primary screen had a Z' = 0.5 and its hit rate was 0.6% (compounds were considered a hit if their luciferase readouts were beyond three standard deviations of the mean luminescence intensity of thapsigargin treated cells, which corresponded to 54% inhibition). Of these, only 187 compounds did not hit a luciferase-based XBP1 splicing reporter used as proxy to measure activation of the IRE1 branch of the UPR. Thus, these were considered unique to the PERK branch and were cherry-picked for further analysis.

## High-content microscopy-based secondary screen

U2OS cells carrying the ATF4-dGFP-IRES-Cherry reporter were plated in 96 well plates and treated with 100 nM Thapsigargin and 10 µM of the cherry-picked compounds for 8 hr. Cells were stained with Hoechst 33,258 and were visualized using an automated microscope (InCell Analyzer 2000; GE Healthcare, Waukesha, WI). Data acquisition and image analyses were performed with the INCell Developer Toolbox Software, version 1.9 (GE Healthcare, Waukesha, WI). Compounds that blocked induction of the ATF4-dGFP reporter, did not block the accumulation of mCherry downstream of the IRES, and were deemed non-toxic as determined by cell number measured by counting nuclei, were repurchased for further analyses.

## Protein analysis

Cells were lysed in SDS-PAGE loading buffer (1% SDS, 62.5 mM Tris-HCl pH 6.8, 10% glycerol). Lysates were sonicated and equal amounts were loaded on SDS-PAGE gels (BioRad, Hercules, CA). Proteins were transferred onto nitrocellulose and probed with primary antibodies diluted in Tris-buffered saline supplemented with 0.1% Tween 20 and 5% bovine serum albumin. The following antibodies were used: CREB-2 (C-20) (1:800), eIF2Bε (B-7) (1/500) (Santa Cruz Biotechnologies, Dallas, TX); PERK (D11A8) (1:1000), PERK (C33E10) (1:1000), eIF2α (9722) (1:1000), phospho-eIF2α (Ser51) (D9G8) XP (3398) (1:1000) (Cell Signaling Technology, Danvers, MA); XBP1s (C-terminus) (1:500) (BioLegend, San Diego, CA); phospho-S539 eIF2Bε (1/1000) (Abcam); M2 Flag (1:1000) (Sigma, St Louis, MO). An HRP-conjugated secondary antibody (Amersham, Piscataway, NJ) was employed to detect immune-reactive bands using enhanced chemiluminescence (SuperSignal; Thermo Scientific, Waltham, MA).

## Immunofluorescence

U2OS cells were seeded on Slide Flasks (Thermo Scientific, Waltham, MA) 18 hr prior to processing for immunofluorescence. Cells (60% confluent) were fixed with 4% paraformaldehyde in PBS for 15 min. The cells were then rinsed three times with PBS and permeabilized with 0.3% Triton X-100. The fixed cells were rinsed three times with PBS and blocked for 1 hr at room temperature in PBS supplemented with 0.1% Triton X-100 and 5% normal goat serum. The cells were then incubated overnight at 4°C with an anti-CHOP mouse monoclonal antibody (Cell Signaling Technology L63F7, Danvers, MA) at a 1:1000 dilution in blocking buffer. The next morning the slides were washed three times (5 min each time) with PBST (PBS-0.1% Triton X-100). The slides were then incubated for 1 hr at room temperature in a 1:500 dilution (in blocking buffer) of secondary anti-mouse antibody labeled with Alexa Dye 488 (Molecular Probes, Invitrogen, Carlsbad, CA). The slides were then washed three additional times with PBST. The cells were then counterstained with rhodamine-phalloidin (1:1000 in PBS) for 10 min at room temperature to reveal the actin cytoskeleton. Lastly, the slides were mounted using Vectashield (Vector, Burlingame, CA) mounting medium and imaged using a Zeiss Axiovert 200M epifluorescence microscope.

## Polysome gradients

Mouse embryonic fibroblasts (MEFs) or TREx-293 cells expressing eIF2α(S51D) were seeded on 150-mm plates and allowed to grow to 80% confluence. Cells were then induced with 25 nM doxycycline for 14 hr and subsequently treated with the appropriate compounds for the indicated times. 100 µg/ml of cycloheximide was added to the cells for 1 min before lysis. Cells were washed twice with PBS supplemented with 100 µg/ml cycloheximide and subsequently lysed in 20 mM Tris pH 7.4, 200 mM NaCl, 15 mM MgCl, 1 mM DTT, 8% Glycerol, 100 µg/ml cycloheximide, 1% Triton X-100 and EDTA-free protease inhibitor tablets (Roche, South San Francisco, CA). Cells were scraped, collected, triturated with a $25^{7/8}$ gauge needle, and the homogenate was centrifuged for 10 min at $10,000 \times g$. The supernatant was loaded on a 10–50% sucrose gradient and sedimented in a SW40 rotor at $150,000 \times g$ for 2.4 hr. The gradients were fractionated using a piston gradient fractionator (BioComp Instruments, Fredericton, NB, Canada) and UV absorbance at 254 nm was monitored using a UV-Monitor (BioRad, Hercules, CA).

## Alpha screen for phospho-S51 eIF2α

U2OS cells were plated on 96-well plates and left to recover overnight. Cells were treated with either with 2 µg/ml tunicamycin or 100 nM thapsigargin in the presence or absence of 100 nM ISRIB or with ISRIB alone for the indicated and the level of eIF2α phosphorylation was determined using the AlphaScreen SureFire eIF2α(p-Ser51) Assay kit (Perkin Elmer, Waltham, MA) following the manufacturer's recommendations. Plates were read in an Envision Xcite Multilabel Reader using the standard Alpha Screen settings.

## Metabolic labeling

HEK293T cells were seeded on 12-well plates, allowed to recover overnight and treated for the indicated times with the indicated compounds. The cells were subsequently switched to media lacking methionine and cysteine supplemented with the indicated compounds and 50 µCi of $^{35}$S-methionine (Perkin Elmer, Waltham, MA) for 20 min. Cells were lysed by addition of SDS-PAGE loading buffer. Lysates were sonicated and equal amounts were loaded on SDS-PAGE gels (BioRad, Hercules, CA). The gel was dried and radioactive methionine incorporation was detected by exposure to a phosphor-screen and visualized with a Typhoon 9400 Variable Mode Imager (GE Healthcare Waukesha, WI).

## Live cell imaging

T-REx293 cells carrying GFP-IRE1 were imaged as described in Li et al., PNAS (*Li et al., 2010*).

## Caspase3/7 activation

Hela cells were plated in 96-well Corning plates at $0.4 \times 10^4$ cells per well, 24 hr prior to imaging. On the day of experiment, DMEM media was replaced with F12 media with appropriate concentration of inhibitors and ER stress inducers and caspase 3/7 reagent at 1:1000 dilution (Essen Bioscience No. 4440, Ann Arbor, MI). Cells were imaged in the IncuCyte FLR live cell imaging system at 2 hr intervals for 70 hr. In order to quantify the total number of cells, Vybrant DyeCycle Green staining solution (1 µM) was added directly to the well immediately after the final Caspase-3/7 scan and incubated for 1 hr prior to acquiring final images. Data was analyzed using IncuCyte analysis software.

## qRT-PCR

U2OS cells were plated on 96-well plates and allowed to recover overnight. Cells were treated for the indicated times with the indicated compounds, lysed and cDNA was synthesized using the PowerSYBR Green Cells-to-CT kit (Ambion, Invitrogen, Carlsbad, CA) following the manufacturer's recommendations. The reactions were ran in an Opticon 2 thermal cycler (BioRad, Hercules, CA) and analyzed with the Opticon Monitor v3 software (BioRad, Hercules, CA). The following oligonucleotides were used for the amplification reaction: Human GADD34: 5'-GTAGCCTGATGGGGTGCTT -3' and 5'- TGAGGCAGCCGG AGATAC -3'; Human CHOP: 5'- AGCCAAAATCAGAGCTGGAA -3' and 5'-TGGATCAGTCTGGAAAA GCA -3'; Human GAPDH: 5'-TGGAAGATGGTGATGGGATT -3' and 5'- AGCCACATCGCTCAGACAC -3'.

## TaqMan assay to measure XBP1 mRNA splicing

cDNA obtained with the PowerSYBR Green Cells-to-CT kit (Ambion, Invitrogen, Carlsbad, CA) as described above was used for the Taqman Assay. TaqMan assays were set up using iQ Supermix (BioRad, Hercules, CA), 250 nM of each outer primer, 200 nM FAM-XBP1U probe, or 100 nM HEX-XBP1S probe. The reactions were then run on a real-time DNA Engine Opticon 2 PCR thermal cycler (BioRad) and analyzed with the Opticon Monitor v3 software (BioRad). The outer primers employed for the human XBP1unspliced/spliced (u/s) TaqMan assay were: 5'-GAAGCCAAGGGGAATGAAGT-3', and 5'-GAGATGTTCTGGAGGGGTGA-3'. TaqMan probes specific for human XBP1s or XBP1u were: 5'-FAM-CAGCACTCAGACTACGTGCACCTCTG-BHQ1-3', and 5'-HEX-TCTGCTGAGTCCGCAGCAGGTGCA-BHQ1-3'.

## RNA isolation and semi-quantitative RT-PCR

Total RNA from treated or untreated HEK293T cells was extracted using TRIzol (Invitrogen, Carlsbad, CA) following the manufacturer's recommendations. 500 ng of total RNA were reverse transcribed using the SuperScriptVilo cDNA Synthesis kit (Invitrogen). The cDNA was diluted 1 in 10 in TE (pH = 8) and 1% of the total reaction was used as a template for the PCR amplification reactions. The XBP1 primers flank the 26-nucleotide intron and produce both spliced (222 bp) and unspliced (248 bp) amplicons. The PCR products were resolved in 2.5% agarose. The following oligonucleotides were used for the amplification reaction: for human XBP1, 5'-ACTGGGTCCAAGTTGTCCAG -3' and 5'- GGAG TTAAGACAGCGCTTGG -3'; for human GAPDH 5'- TGGAAGATGGTGATGGGATT -3' and 5'-AGCCACA TCGCTCAGACAC -3'.

## Pharmacokinetics of ISRIB

Intra-peritoneal (ip) route of administration was performed on 6–7 wk old female CD-1 mice (Harlan Laboratories, Indianapolis, IN). Animals received a single, 5 mg/kg dose in groups of three mice/compound/route of administration. ISRIB was dissolved in DMSO then diluted 1:1 in Super-Refined PEG 400 (Croda, Edison, NJ). Blood (80 µl) was collected from the saphenous vein at intervals post-dosing

(20 min, 1 hr, 3 hr, 8 hr, 24 hr) in EDTA containing collection tubes (Sarstadt CB300) and plasma was prepared for analysis. Compounds were detected by time-of-flight mass spectroscopy.

Intra-peritoneal (ip) route of administration was performed at a single dose of 2.5 mg/kg in groups of three for each time-point (2, 6, 24 and 36 hr). Brain tissue samples were individually homogenized with a Tissue Tearor (Model 985-370 type2, BioSpec Products Inc, Bartlesville, OK). Approximately 300 mg of tissue was placed in 5-ml polypropylene tube, and four volumes of water were then added to mix. The speed scale of Tissue Tearor was set at 3 for 2 min. After homogenization, the supernatant was analyzed by LC-MS/MS to determine their brain concentration. Plasma samples were collected prior to extraction of brain samples.

## Memory studies
Eight to ten-week-old male C57BL/6J mice were used for behavioral experiments. Food and water were provided ad libitum, and mice were kept on a 12:12 hr light/dark cycle (lights on at 08:00 hr). All procedures complied with Canadian Council on Animal Care guidelines.

## Morris water maze
Mice were trained in a water pool of 100 cm diameter with a hidden platform of 10 cm diameter. Mice were handled daily for 3 days before the experiment, and the training protocol consisted of one swimming trial per day. Each mouse swam until it found the hidden platform or 120 s, when it was gently guided to the platform and stayed there for 10 s before being returned to the cage. Immediately after the swimming trial the mice were injected intraperitoneally with ISRIB (0.25 mg/kg in saline, 1% DMSO). For the probe test, the platform was removed and each mouse was allowed to swim for 60 s, while its swimming trajectory was monitored with a video tracking system (HVS Image, Buckingham).

## Contextual fear conditioning
Mice were trained with a protocol that consisted of a 2-min period of context exploration, followed by a single foot shock of 0.35 mA for 1 s. Mice received a single injection of ISRIB (2.5 mg/kg in 50% DMSO, 50% PEG 400, IP) immediately after training and were returned to their home cage. One and 24 hr after training, the mice were tested for contextual fear memory by placing the animals in the conditioning context for a 4-min period. The incidence of freezing was scored in 5-s intervals as either 'freezing' or 'not freezing'. Percent of freezing indicates the number of intervals in which freezing was observed divided by total number of 5-s intervals. Statistical analyses were done by Student's *t* tests and one-way ANOVA followed by between-group comparisons using Tukey's posthoc test.

## Cannulation and auditory fear conditioning
Male Sprague Dawley rats (275–350 g) were used for cannulation as described in Migues et al., 2010 (*Migues et al., 2010*). ISRIB (0.05 mg/ml, 0.5 µl) was infused bilaterally into the amygdala immediately after auditory fear conditioning training. The infusion was performed with a microinjector (28 gauge) connected to a Hamilton syringe with plastic tubing at a rate of 0.25 µl/min. To allow for the solution containing ISRIB to diffuse from the tip of the cannula into the tissue, the microinjector stayed in the cannula for one additional minute. Training protocol for auditory fear conditioning consisted of a 2-min period of context A exploration, followed by one pairing of a tone (5000 Hz, 75 dB, 30 s) with a co-terminating foot shock (0.75 mA, 1 s). Rats were returned to their home cage 1 min after the shock. Test for auditory fear memory consisted of a 2 min acclimatizing period to the context B (pre-CS), followed by tone presentation (CS) (2800 Hz, 85 dB, 30 s). Freezing time was measured and percent of freezing was calculated. At the end of the experiment, cannula placement was checked by examining 50 µm brain sections stained with formal-thionin under a light microscope.

## Synthesis of ISRIB
### General methods
Commercially-available reagents and solvents were used as received. Silica gel chromatography was performed using a Biotage Isolera Four flash purification system with Silicycle Silia*Sep* cartridges. [1]H NMR spectra were recorded on a Varian INOVA-400 400 MHz spectrometer. Chemical shifts are reported in δ units (ppm) relative to residual solvent peak. Coupling constants (*J*) are reported in hertz (Hz). LC-MS analyses were carried out using a Waters 2795 separations module equipped with a Waters 2996 photodiode array detector, a Waters 2424 ELS detector, a Waters micromass ZQ single quadropole mass detector, and an XBridge C18 column (5 µm, 4.6 x 50 mm).

## Synthesis of bisglycolamides

trans-ISRIB: 2-(4-Chlorophenoxy)-N-[(1r,4r)-4-[2-(4-chlorophenoxy)acetamido] cyclohexyl] acetamide
To a mixture of (1r,4r)-cyclohexane-1,4-diamine (20 mg, 0.18 mmol) in tetrahydrofuran:water (1:1, 1 ml) were sequentially added potassium carbonate (73 mg, 0.53 mmol) and 4-chlorophenoxyacetyl chloride (56 μl, 0.36 mmol). Upon addition of the acid chloride, a white solid immediately formed. The reaction mixture was vigorously stirred at ambient temperature for 30 min. Water (2.5 ml) was added. The mixture was vigorously vortexed then centrifuged, and the water was decanted. This washing protocol was repeated with potassium bisulfate (1% aq, 2.5 ml), water (2.5 ml), and diethyl ether (2 × 2.5 ml). The resulting wet white solid was dried by partially dissolving in dichloromethane/methanol (10/1, 10 ml) and gravity filtering through an Autochem 4.5-ml reaction tube. The residual undissolved product was extracted from the wet filter cake by adding dichloromethane (4 × 4.5 ml) and gravity filtering. The combined filtrate was concentrated using rotary evaporation to afford 51 mg (65%) of the title compound as a white solid. $^1$H NMR (400 MHz, DMSO-d$_6$) δ 7.91 (d, $J$ = 8.1 Hz, 2H), 7.31 (d, $J$ = 9.0 Hz, 4H), 6.94 (d, $J$ = 9.0 Hz, 4H), 4.42 (s, 4H), 3.55 (br. s., 2H), 1.73 (br. d, $J$ = 5.9 Hz, 4H), 1.30 (quin, $J$ = 10.5 Hz, 4H); LC-MS: $m/z$ = 451 [M+H, $^{35}$Cl x 2]$^+$, 453 [M+H, $^{35}$Cl, $^{37}$Cl]$^+$.

cis-ISRIB: 2-(4-chlorophenoxy)-N-[(1s,4s)-4-[2-(4-chlorophenoxy) acetamido] cyclohexyl]acetamide
To a mixture of (1s,4s)-cyclohexane-1,4-diamine (21 μl, 20 mg, 0.18 mmol) in tetrahydrofuran:water (1:1, 1 ml) were sequentially added potassium carbonate (73 mg, 0.53 mmol) and 4-chlorophenoxyacetyl chloride (56 μl, 0.36 mmol). The reaction mixture was vigorously stirred at ambient temperature for 1.5 hr then partitioned between 30 ml of 1:1 dichloromethane:KHSO$_4$ (10% aq.). After separating the organic layer, it was sequentially washed with water (1 × 10 ml) and brine (1 × 10 ml) then dried by gravity filtration using an Autochem 4.5-ml reaction tube. The filtrate was concentrated and loaded onto a Silicycle 4g SiO$_2$ column using a minimal amount of dichloromethane (~2 ml). The product was eluted with acetone in dichloromethane (0–50%). Product-containing fractions were combined and concentrated to afford 56 mg (71%) of the title compound as a white solid. $^1$H NMR (400 MHz, DMSO-d$_6$) δ 7.76 (d, $J$ = 7.0 Hz, 2H), 7.32 (d, $J$ = 9.0 Hz, 4H), 6.94 (d, $J$ = 9.0 Hz, 4H), 4.47 (s, 4H), 3.70 (br. s., 2H), 1.44 − 1.67 (m, 8H); LC-MS: $m/z$ = 451 [M + H, $^{35}$Cl x 2]$^+$, 453 [M + H, $^{35}$Cl, $^{37}$Cl]$^+$.

## Acknowledgements

We thank Yong Huang and Blake Aftab for mass spectrometry and pharmacokinetic analysis and the members of the Walter lab for critical reading of the manuscript. We thank Paige Nittler for her help coordinating this collaboration, and Marc Shuman for his invaluable advice and guidance. This work was funded through an HHMI Collaborative Innovation Award. We thank the HHMI leadership for establishing this visionary funding mechanism, without which this work would not have been possible. PW is an Investigator of the Howard Hughes Medical Institute.

# Additional information

## Competing interests

NS: Reviewing Editor, *eLife*. The other authors declare that no competing interests exist.

## Funding

| Funder | Grant reference number | Author |
|---|---|---|
| Howard Hughes Medical Institute Collaborative Innovation Award | | Carmela Sidrauski, Diego Acosta-Alvear, Punitha Vedantham, Brian R Hearn, Ciara Gallagher, Chris Wilson, Voytek Okreglak, Byron Hann, Michelle R Arkin, Adam R Renslo, Peter Walter |
| Canadian Institutes of Health Research | (N.S., MOP-114994) | Arkady Khoutorsky, Karine Gamache, Karim Nader, Nahum Sonenberg |
| Irvington Institute Postdoctoral Fellowship of the *Cancer* Research Institute | | Diego Acosta-Alvear |
| QB3-Malaysia Program | | Kenny K-H Ang |

The funders had no role in study design, data collection and interpretation, or the decision to submit the work for publication.

## Author contributions

CS, Conception and design, acquisition of data, analysis and interpretation of the data, writing the manuscript; DA-A, Conception, design and construction of reporters, analysis of data and revising the article; AK, NS, Memory studies conception and design, acquisition and analysis of data; PV, BRH, ARR, Chemistry design and analysis of data; HL, AA, Design, acquisition and analysis of caspase 3/7 data; KG, KN, Fear conditioning experiments in rats; CG, Construction and characterization of tagged ATF6 cell line, data acquisition; KK-HA, CW, MRA, High throughput screening and analysis of screening data; VO, High throughput screening; BH, Design of pharmacokinetic studies and interpretation of data; PW, Conception and design, data analysis and interpretation of the data, writing the manuscript

## Ethics

Animal experimentation: All procedures complied with Canadian Council on Animal Care guidelines. Animal Use protocol # 4512 (McGill University); Animal Use protocols # 5329 and 5205 (McGill University).

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
