## [Decision Letter]

Thank you for choosing to send your work entitled “Pharmacological brake-release of mRNA translation enhances cognitive memory”for consideration at *eLife*. Your article has been evaluated by a Senior editor and 4 reviewers, one of whom is a member of our Board of Reviewing Editors.

The Reviewing editor and the other reviewers discussed their comments before we reached this decision, and the Reviewing editor has assembled the following comments to help you prepare a revised submission.

Our consensus is that the discovery of a small molecule that interposes itself (functionally) between the phosphorylated alpha subunit of translation initiation factor 2 (eIF2α) and the target of this phosphorylation event, the translational machinery, thereby inhibiting an ancient signaling pathway known as the integrated stress response, is an important discovery. No one has previously reported a functionally similar inhibitor–in fact, until this report, it was unsuspected that this node in the ISR was even susceptible to inhibition.

The compound is remarkably potent, hitting its unknown target(s) when applied to cells at a submicromolar concentration.

The functional characterization of the compound is rather thorough and thus its utility as a tool compound is well documented–this is important, as many other ‘firsts’ of this type report on theoretical tool compounds, but fall short of demonstrating in vivo utility. Thus, there is reason to hope that ISRIB will be applied to much further research and that this paper will serve as a foundation for such studies.

Lastly, the reviewers were impressed by effects on memory formation, which is enhanced by application of the compound.

Whereas the functional characterisation of ISRIB is very thorough, it is not matched by a similar level of molecular detail. We do not know if ISRIB directly modulates the interaction of eIF2α with eIF2B or the competing interactions with eIF5 or some aspect of eIF2B activity–possibilities noted by the authors in their Discussion.

The four reviewers were initially divided on the impact of this deficiency on the suitability of the paper for publication in *eLife*. The conclusion of our deliberations is that some simple measures–that will provide further clues to the site of the compound's action–need to be taken before the paper will be acceptable for publication in *eLife*.

1) The effect of ISRIB application on the recovery of a complex between eIF2 and eIF2B could be measured directly in lysates of unstressed and stressed cells by co-immunoprecipitation. Chris Proud’s lab has expressed tagged versions of eIF2B subunits that associate with endogenous eIF2 in transfected HEK293 cells (they co-precipitate eIF2α and eIF2α-P by pulling-down the tagged eIF2B subunit (see Figure 2 in PMID: 21560189). If a coherent effect is observed (say less eIF2α-P associating with tagged eIF2Be in ISRIB-treated stressed cells) the discussion, focusing on that aspect, would be justified. However, if no effect is observed, the alternative discussion (focusing on a possible enhancement of eIF2B’s enzymatic activity) would be justified.

2) Similarly, the impact of ISRIB application on those aspects of the phosphorylation status of eIF2B that can be queried by commercially available phospho-specific antibodies should be measured.

3) In relation to the compound’s effects on memory, it was deemed important to assess the likelihood that ISRIB affects memory formation by acting on brain cells, as opposed to eliciting these effects by peripheral action. Therefore it would seem important to determine if the compound gets into the CNS, when applied at doses that affect memory. Perhaps it is possible to analyze levels in the CSF of injected animals or address this matter functionally by measuring ISR activity in the CNS of injected animals in a suitable experimental design.

4) Finally ISRIB treatment was post-training for water maze and cued fear conditioning (in order to focus on effects of ISRIB on memory consolidation), but ISRIB treatment was chronic prior to training for contextual fear conditioning. The authors are asked to address how this affects the conclusions regarding the compounds impact on memory consolidation.

---

## [Author Response]

*1) The effect of ISRIB application on the recovery of a complex between eIF2 and eIF2B could be measured directly in lysates of unstressed and stressed cells by co-immunoprecipitation. Chris Proud’s lab has expressed tagged versions of eIF2B subunits that associate with endogenous eIF2 in transfected HEK293 cells (they co-precipitate eIF2α and eIF2α-P by pulling-down the tagged eIF2B subunit (see Figure 2 in PMID: 21560189). If a coherent effect is observed (say less eIF2α-P associating with tagged eIF2Be in ISRIB-treated stressed cells) the discussion, focusing on that aspect, would be justified. However, if no effect is observed, the alternative discussion (focusing on a possible enhancement of eIF2B’s enzymatic activity) would be justified*.

We have performed the experiment suggested (Chris Proud kindly provided the plasmids encoding the myc-tagged subunits of eIF2B) and we are able to co- immunoprecipitate eIF2α after a Ni-NTA pull-down. However, no differences are observed in the amount associated of either the phospho- or total eIF2α in the presence or absence of ISRIB. We have not pursued this effort further because we are overexpressing the GEF significantly, and we believe that the absence of effect of ISRIB on the interaction between eIF2B and eIF2 may not necessarily reflect what happens under physiological expression levels.

In addition, we have performed sucrose gradients to reveal changes in the interaction of eIF2 and eIF2B following the endogenous complexes (HEK293T cells) with antibodies against eIF2α and eIF2Bε. We do not observe significant changes in complex formation (eIF2B-eIF2) upon ER-stress and thus we are not able to test the effect of ISRIB on this interaction.

We are currently developing potent ISRIB analogs that will allow us to cross-link ISRIB to its target. Our strategy is to develop appropriate biochemical assays to unravel the molecular mechanism after identification of its target. Although increasing eIF2B activity or decreasing the interaction of phospho- eIF2α with eIF2B are possible modes in which ISRIB could generate a resistant phenotype, there are many other ways in which this molecule could act. For example ISRIB could modulate the interaction between eIF2 and eIF5, which acts as a GAP and GDI. We believe that the unbiased search for ISRIB’s target will be a more linear path to unravel its mechanism of action than ruling in or out individual possibilities.

*2) Similarly, the impact of ISRIB application on those aspects of the phosphorylation status of eIF2B that can be queried by commercially available phospho-specific antibodies should be measured*.

We queried the phosphorylation status of Serine 539 of the eIF2Bε subunit and found no differences in the presence or absence of ISRIB. This site is constitutively phosphorylated by GSK and negative regulates eIF2B activity. In macrophages, dephosphorylation of S539 is induced upon pathogen invasion and TLR engagement, leading to increased eIF2B GEF activity inducing a phospho-eIF2α resistant phenotype (Woo et al, Nat. Cell Biol, January 2012). ISRIB does not elicit this phenotype by modulating this phosphorylation site. We added the data to Figure 3—figure supplement 1 and mentioned this result in the Discussion. To our knowledge no other phospho-specific antibodies are commercially available for eIF2B putative regulatory phosphorylation sites.

*3) In relation to the compound’s effects on memory, it was deemed important to assess the likelihood that ISRIB affects memory formation by acting on brain cells, as opposed to eliciting these effects by peripheral action. Therefore it would seem important to determine if the compound gets into the CNS, when applied at doses that affect memory. Perhaps it is possible to analyze levels in the CSF of injected animals or address this matter functionally by measuring ISR activity in the CNS of injected animals in a suitable experimental design*.

We have confirmed the presence of ISRIB in the brain by mass spectrometry. We performed a single intraperitonneal injection at 2.5 mg/kg (the dose used in the context-cued fear conditioning experiment) and detect the appearance of ISRIB in the brain with the same dynamics of accumulation that we observed in plasma. After 24 hr, the time-point at which long-term memory is measured, the concentration of ISRIB in the brain is 30 ng/g of tissue or approximately 60 nM (12-fold higher than its IC_50_). We have included the pharmacokinetic data in Figure 6.

We have not been able to look at ISR activity in the brain in the experimental setup that replicates the behavioral tests. In this scenario, there is no significant result that increases eIF2α phosphorylation or ATF4 production. We postulate that memory depends on transient and local changes in eIF2α phosphorylation. Looking at the modulation of eIF2α phosphorylation in the brain and the effect of ISRIB will have to await the development of more sensitive in vivo tools that report on this molecular event, such as an ATF4-luciferase reporter mouse.

*4) Finally ISRIB treatment was post-training for water maze and cued fear conditioning (in order to focus on effects of ISRIB on memory consolidation), but ISRIB treatment was chronic prior to training for contextual fear conditioning. The authors are asked to address how this affects the conclusions regarding the compounds impact on memory consolidation*.

The cued fear conditioning experiment was repeated and ISRIB was injected only once after training at a higher concentration (2.5 mg/kg instead of 0.25 mg/kg).

We observed increased freezing with this new dosing and have changed the data in Figure 6 and have updated the Materials and methods section.